# Medical Ozone Treatment Attenuates Male Reproductive Toxicity Induced by Bleomycin, Etoposide, and Cisplatin Regimen in an Experimental Animal Model

**DOI:** 10.3390/ijms26178547

**Published:** 2025-09-03

**Authors:** Necdet Altıner, Yaprak Dönmez Çakıl, İdil Duran, Damla Gökçeoğlu Kayalı, Hale Bayram, Abdullah Pehlivan, Oğuz Kaan Tombul, Belgin Selam, Mehmet Cıncık, Mustafa Erinç Sitar

**Affiliations:** 1Department of Histology and Embryology, Faculty of Medicine, Maltepe University, 34858 Istanbul, Türkiye; necdetaltiner@maltepe.edu.tr (N.A.); halebayram@maltepe.edu.tr (H.B.); mehmet.cincik@maltepe.edu.tr (M.C.); 2Experimental Animals Research and Application Center, Maltepe University, 34858 Istanbul, Türkiye; oguzkaan.tombul@maltepe.edu.tr (O.K.T.); merincsitar@maltepe.edu.tr (M.E.S.); 3Department of Medical Biology and Genetics, Faculty of Medicine, Maltepe University, 34858 Istanbul, Türkiye; yaprak.cakil@maltepe.edu.tr; 4Department of Obstetrics and Gynecology, Faculty of Medicine, Maltepe University, 34858 Istanbul, Türkiye; dr.idilduran@gmail.com; 5Histology and Embryology Department, Marmara University School of Medicine, 34854 Istanbul, Türkiye; damlagokceoglu@yandex.com; 6Faculty of Medicine, Maltepe University, 34857 Istanbul, Türkiye; abdulllahp94@gmail.com; 7Department of Obstetrics and Gynecology, School of Medicine, Acibadem Mehmet Ali Aydinlar University, 34638 Istanbul, Türkiye; 8Department of Clinical Biochemistry, Faculty of Medicine, Maltepe University, 34858 Istanbul, Türkiye

**Keywords:** bleomycin, cancer, chemotherapy, cisplatin, etoposide, infertility, medical ozone, oxidative stress, rat, sperm

## Abstract

The chemotherapeutic combination of bleomycin, etoposide, and cisplatin (BEP) is well-documented to exert gonadotoxic effects, ultimately leading to impaired fertility. This experimental rat study investigated the potential protective role of repeated medical ozone therapy in mitigating the deleterious effects of BEP treatment in male rats. Thirty-two adult male Sprague Dawley rats were randomly assigned to four groups: (i) a healthy control group, (ii) a group receiving injections of the BEP regimen over nine weeks, (iii) a group receiving the same BEP regimen plus medical ozone (1 mg/kg IP) twice weekly, and (iv) a group receiving only ozone therapy. BEP treatment significantly reduced sperm concentration and increased morphological abnormalities, both of which were partially restored by ozone co-administration. Ozone therapy also elevated testosterone and thyroid-stimulating hormone (TSH) levels when co-administered with BEP compared to BEP treatment alone. Oxidative stress analysis demonstrated that total oxidative status (TOS) and total antioxidant status (TAS) levels were significantly improved in the BEP + ozone group. Histopathological analysis revealed that ozone treatment ameliorated BEP-induced testicular damage, as evidenced by improved Johnsen scores and increased thickness of the seminiferous tubule epithelium. In conclusion, repeated medical ozone therapy appears to mitigate BEP-induced reproductive toxicity by preserving sperm quality, endocrine function, and redox homeostasis.

## 1. Introduction

Cancer is the second leading cause of death worldwide. Chemotherapeutic agents represent the primary treatment option for a wide variety of malignancies, including but not limited to breast, thyroid, ovarian, leukemia, lung, malignant lymphoma, multiple myeloma, and testicular cancer [1,2,3]. Among the available treatment options, the combination of bleomycin, etoposide, and cisplatin (BEP) is a significant protocol that has been shown to result in a five-year survival rate of 90% in the treatment of testicular cancer [4,5]. However, due to the lack of selective toxicity, they can damage a wide range of healthy tissues, including those of the gastrointestinal, respiratory, and reproductive systems [6,7,8]. Specifically, the BEP regimen has been demonstrated to induce severe gonadotoxic effects, with a significant impact on fertility in both the short and long term [2,9,10,11]. Preclinical studies have demonstrated that BEP treatment can result in sperm head protein disruptions, spermatogenic cell apoptosis, and seminiferous tubular atrophy, eventually leading to sperm count and motility abnormalities [12,13,14,15]. Furthermore, BEP treatment has been shown to increase the expression of oxidative stress as well as apoptosis-related genes, leading to chromatin disruption and DNA methylation alterations in spermatogonial stem cells [15,16]. BEP has also been demonstrated to impair Leydig cell function, thereby contributing to testicular endocrine disruption [17].

In order to mitigate or prevent the gonadotoxic effects of BEP, several antioxidants have been administered in various in vitro and in vivo studies. Antioxidant treatments have been shown to alleviate reproductive system damage by improving various parameters, including sperm count, serum testosterone levels, and testicular histopathology [2,13,14,18,19]. Medical ozone (O_3_) is a potent oxidizing agent utilized in medical applications at low and controlled doses. Following administration, ozone rapidly dissolves in the body, generating reactive oxygen species (ROS) and lipid peroxidation products (LOPs). These molecules create a controlled oxidative stress environment within cells, thereby stimulating antioxidant defense mechanisms. Consequently, cells exhibit increased resistance to oxidative stress, while molecular responses essential for tissue repair and inflammation regulation are also activated [20,21,22]. Medical ozone therapy induces the activities of key antioxidant enzymes, including catalase (CAT), superoxide dismutase (SOD), and glutathione peroxidase (GSH-Px) [3,23,24,25]. This antioxidant potential renders medical ozone a promising therapeutic approach, particularly in conditions associated with oxidative damage, such as male infertility. Ozone therapy can improve testicular function, spermatogenesis, and various sperm parameters by reducing oxidative stress in testicular tissue [3,25,26]. Ozone has therapeutic, non-effective, and toxic concentration ranges, and its safety depends on the dose and administration method. The Madrid Declaration on Ozone Therapy (2010) defines disease-specific dosage ranges and routes, noting that ozone concentrations as low as 5–10 µg/mL, and even lower, can exert therapeutic effects while maintaining a wide safety margin [27]. Similarly, Serra et al. (2023) mapped evidence across medical conditions and administration routes, confirming that therapeutic ozone has a wide safety margin and does not exhibit toxicity within this range [28]. Nevertheless, the optimal dosage, administration route, and treatment duration of ozone therapy vary widely across studies and remain to be standardized for different pathological conditions. Moreover, to our knowledge, no study has investigated the possible protective role of ozone against BEP-induced gonadotoxicity.

This study aims to demonstrate whether medical ozone therapy can attenuate the pro-oxidative, endocrine-disrupting, and gonadotoxic effects of BEP chemotherapy in a rat model.

## 2. Results

### 2.1. Effect of Ozone on Body Weight and Testis Weight Indices

As shown in Figure 1, BEP treatment significantly reduced the body weight gain (*p* = 0.0001) and testicular weight index (TWI) (*p* = 0.0004) in comparison with the control group. Ozone co-treatment significantly improved the body weight gain when compared to BEP treatment alone (*p* = 0.0002); however, it did not affect TWI, with no significant difference observed between the respective groups (*p* = 0.999). Ozone treatment alone resulted in a lower weight gain in comparison with the control group (*p* = 0.0097), but no significant difference in TWI was found between the control and ozone groups.

### 2.2. Effect of Ozone on Sperm Parameters

As presented in Figure 2A, BEP treatment resulted in a significantly decreased sperm concentration relative to the control group (*p* = 0.0118), which was partially restored in the BEP and ozone therapy group (*p* = 0.0001). Ozone treatment alone did not cause a significant difference when compared to the control group (*p* = 0.1426).

BEP treatment significantly increased the number of sperm exhibiting morphological abnormalities in the head, neck, and tail regions, as well as multiple anomalies when compared to the control group (*p* = 0.0001 for all parameters, Figure 2B–F, Appendix A). When ozone therapy was applied together with BEP treatment, ozone significantly reduced the number of sperm with head, neck, and tail abnormalities (*p* = 0.0001 for all parameters, Figure 2C–E). The sperm counts with multiple abnormalities (Figure 2F) were similar between the two groups (*p* = 0.099).

Interestingly, the group that received only ozone therapy displayed a significantly higher number of sperm with head (*p* = 0.0421) and tail (*p* = 0.0001) abnormalities compared to the control group (Figure 2C,E). No significant difference was observed in the number of sperm with neck and multiple anomalies between the control and ozone therapy groups (*p* = 0.084 and 0.591, respectively, Figure 2D,F).

As depicted in Figure 3, BEP treatment was found to significantly decrease sperm viability compared to the control group (*p* = 0.0001), which was restored by ozone co-treatment (*p* = 0.0001 in comparison with the BEP treatment alone). The group that received only ozone therapy also had a slightly lower percentage of sperm viability compared to the control group (*p* = 0.0249).

### 2.3. Effect of Ozone on Testicular Histopathology

As demonstrated in Figure 4, histopathological analysis following BEP treatment revealed characteristic testicular damage, including thinning of the seminiferous epithelium, the presence of vacuoles within the epithelium, loss of interstitial space integrity, and desquamation of germ cells. These pathological alterations were partially alleviated in the group receiving co-administration of medical ozone. Histopathological scoring confirmed these findings, showing that BEP treatment significantly reduced epithelial thickness and Johnsen score compared to the control group (*p* = 0.021, *p* = 0.0001, respectively, Figure 4). Ozone co-treatment, on the other hand, increased both epithelial thickness and Johnsen score (*p* = 0.0136, *p* = 0.0005, respectively). Ozone therapy alone did not have a significant effect on epithelial thickness or Johnson score compared to the control group (*p* = 0.99, *p* = 0.9975, respectively).

### 2.4. Effect of Ozone on Redox Status and Endocrine Parameters

TAS, testosterone, and TSH were significantly lower, and TOS was significantly higher in the BEP group compared to those in the control group (*p* = 0.0176, *p* = 0.0001, *p* = 0.0001, *p* = 0.0009, respectively, Figure 5A–D). Ozone therapy markedly elevated testosterone and TSH levels when co-administered with BEP compared to BEP treatment alone (*p* = 0.044, *p* = 0.0044, respectively). Moreover, TOS and TAS levels were partially restored to control levels following the co-administration of the BEP regimen and ozone (*p* = 0.007 and *p* = 0.0182, respectively, in comparison with those in the BEP group).

TAS significantly increased and testosterone significantly decreased in the ozone therapy group relative to those in the control group (*p* = 0.0026 and *p* = 0.024, respectively). TOS and TSH did not differ significantly between these two groups (*p* = 0.933, *p* = 0.717, respectively).

## 3. Discussion

We examined the effects of BEP chemotherapy on the male reproductive function of sexually mature rats in the current study and evaluated whether administration of medical ozone could mitigate its gonadotoxic and pro-oxidative consequences. To our knowledge, this is the first experimental study administering ozone therapy concurrently with the entire 9-week BEP protocol, thereby providing a translationally relevant model of clinical exposure.

BEP administration exhibited a significant decrease in weight gain and testicular index, with severe impairment of testicular histology and spermatogenesis leading to a reduction in sperm count, viability, and an increase in morphological abnormalities [13,14,15,29]. Cisplatin-based chemotherapy induces significant systemic catabolism, as reflected by reduced body weight gain, alongside testicular atrophy and impaired reproductive organ integrity in animal models [30,31]. Three to four cycles of BEP chemotherapy lead to a marked decline in spermatogenesis in male patients, while the probability of achieving normal sperm counts ranges from 22 to 58 percent within a period of two to five years. Furthermore, elevated chemotherapy doses, specifically those involving elevated total dose cisplatin, are associated with a higher incidence of persistent azoospermia in clinical settings [32].

Oxidative stress plays an imperative role in the pathogenesis of gonadotoxicity caused by cisplatin-based chemotherapies. BEP regimen provokes excessive free radical production, enhancing lipid peroxidation and protein carbonyl content in testicular tissue [10,33]. Moreover, DNA breaks, disruption of chromatin integrity, and DNA methylation alterations in spermatozoa are frequently observed following BEP chemotherapy [34,35,36]. Additionally, oxidative stress–induced Sertoli and Leydig cell dysfunction further disrupts the hormonal axes essential for spermatogenesis [12,37].

Recently, medical ozone has received attention as a redox bioregulator in the field of preventive medicine [38]. Mild or low-dose medical ozone therapy exerts its molecular effects primarily through controlled oxidative stress that activates adaptive cellular responses. This oxidative stimulus enhances the antioxidant defense system by upregulating key pathways such as nuclear factor erythroid 2-related factor 2/Kelch-like ECH-associated protein (Nrf2/Keap1), leading to increased expression of phase II detoxifying enzymes and anti-oxidant molecules, including SOD, CAT, and GSH-Px [39]. Additionally, ozone-induced mild oxidative stress promotes the release of growth factors and modulates inflammatory signaling by activating Nrf2, while concurrently inhibiting NF-kB, thereby reducing the production of pro-inflammatory cytokines and favoring an anti-inflammatory phenotype [40]. When activated under oxidative stress, Nrf2 not only enhances antioxidant and anti-inflammatory defenses but also confers antiapoptotic effects and supports mitochondrial function and cellular bioenergetics [22,41]. Collectively, these mechanisms suggest that controlled ozone exposure triggers adaptive responses that restore redox homeostasis and support tissue repair.

Although medical ozone exhibits protective effects at therapeutic doses, it is important to note that higher concentrations may potentially exert toxic effects, highlighting the need for careful dose selection [38]. An ozone dose of 1 mg/kg was selected based on extensive literature evidence and established guidelines [27,42,43]. Additionally, a group receiving ozone alone was included to monitor for any toxic effects.

Our findings demonstrated that repeated medical ozone therapy mitigated BEP-induced reproductive toxicity by preserving sperm quality, endocrine function, and redox homeostasis, as evidenced by partially improved sperm parameters, histological parameters, antioxidant capacity, TSH, and testosterone levels. Although ozone did not completely restore sperm viability, its administration nearly doubled sperm concentration compared to the BEP group. Ozone co-treatment also markedly improved both the Johnsen score and mean seminiferous tubule epithelial thicknesses in rats receiving BEP treatment. While not equivalent to untreated controls, the BEP + ozone group exhibited enhanced testis histology indicative of the successful restoration of spermatogenesis. Nevertheless, the slight increase in sperm abnormalities observed in the ozone-only group indicates that a toxic dose may vary in healthy animals and that optimal dosing differs across disease models. Its application in healthy rats without underlying oxidative stress may lead to an imbalance between ROS and antioxidant defenses. This imbalance can result in oxidative damage to spermatozoa, manifesting as morphological abnormalities. Previous studies have reported that exposure to environmental ozone adversely affects semen quality, including sperm morphology, in humans. It has been demonstrated that the disruption of the antioxidant-oxidant balance in favor of oxidants is a prerequisite for the efficacy of damage prevention [44,45,46]. Moreover, co-treatment with ozone reversed central and gonadal endocrine dysfunction caused by BEP treatment to a notable degree, suggesting preservation of Leydig cell activity and hypothalamic–pituitary axis integrity. However, complete hormonal recovery was not achieved, warranting further investigation with longer-term or different doses of medical ozone application on the endocrine system. At the redox level, BEP elevated TOS and suppressed TAS, consistent with its known role in promoting oxidative stress [16]. The administration of ozone therapy resulted in the partial reversal of this imbalance by means of enhancing the antioxidant capacity.

Our findings are consistent with those of the study by Tusat et al., which demonstrated that ozone therapy can reduce testicular damage caused by ischemia–reperfusion [46]. Moghadam et al. (2021) conducted a study to investigate the effects of ozone and melatonin on busulfan-induced testicular damage in mice. Both applications significantly contributed to the preservation of healthy sperm [26]. In addition, Salem et al. (2017) and Aydoğdu et al. (2019) demonstrated that ozone therapy effectively safeguarded testosterone and TSH levels by impeding the adverse impact of BEP [3,19]. Kheirouri et al. demonstrated that ozone administration facilitates the restoration of TAS levels compared to those observed in the control group [47]. Moreover, the protective effect of medical ozone has been documented in other organs, including the heart. Gülcan et al. (2024) investigated how ozone therapy influenced myocardial ischemia–reperfusion injury in streptozotocin-induced diabetic rats. The researchers reported a reduction in myocardial damage, hemorrhage, and neutrophil infiltration [48].

While the BEP dose used aligns with previous toxicology models [13], there is considerable variation in ozone dosing protocols across the literature. In this study, the lack of complete restoration to control levels in some parameters might be due to the variation of the dosage, administration route, and treatment duration of ozone therapy in comparison with the current literature. Moreover, it is important to note that our findings were obtained in a rat model with IP ozone administration, and the pharmacokinetics, tissue distribution, and physiological responses may differ substantially in humans. In clinical practice, ozone is administered via the intravenous route or via local administration, which may affect its systemic and local effects. Further limitations include that infertile animals and animals with neoplasia were not represented in the study population. Therefore, while our results provide mechanistic insights into the protective potential of ozone against chemotherapy-induced oxidative damage, further studies are necessary to determine optimal dosing, administration routes, and safety profiles for translational application in humans.

## 4. Materials and Methods

### 4.1. Animals and Experimental Design

Adult male Sprague Dawley rats (12–14 weeks old, weighing 250 ± 50 g) were supplied by the Maltepe University Animal Research Center, Istanbul, Türkiye. The rats were maintained under the appropriate environmental conditions with a humidity level ranging from 45% to 55%, a temperature of 18 °C to 24 °C, and a 12 h light/dark cycle. The animals had unrestricted access to tap water and a pellet food diet. The experimental protocol adhered to international guidelines for the care and use of laboratory animals and was approved by the Maltepe University Local Ethics Committee for Animal Experiments (protocol number 2022.09.01). Two rats from the group receiving combined BEP and ozone therapy were excluded from the study in the third week of the first cycle due to excessive weight loss and reduced quality of life, based on the recommendations of the attending veterinarian and the animal welfare committee. These animals were excluded from statistical analyses for all parameters.

### 4.2. Experimental Protocols for the BEP Regimen and Ozone Therapy

The rats were randomly divided into four groups, with eight rats in each group: (i) healthy control, (ii) BEP chemotherapy, (iii) BEP and ozone therapy, and (iv) ozone therapy only (Figure 6). An experimental protocol mimicking “three cycles of 21 days of BEP treatment in humans” was administered to rats over nine weeks. Cisplatin (0.17 mg/kg, CAS: 15663-27-1; Koçakfarma, Istanbul, Türkiye) and etoposide (0.83 mg/kg, CAS: 33419-42-0, Koçakfarma, Istanbul, Türkiye) were administered on days 1–5, while bleomycin (0.083 mg/kg, CAS: 11056-06-7, Koçakfarma, Istanbul, Türkiye) was given on days 2, 9, and 16 of each cycle by intraperitoneal (IP) injection with a 30 min interval between injections [13]. Concurrently, the control group received a 0.9% sodium chloride solution (NaCl). The BEP and medical ozone group was administered with medical ozone (Has Medical^®^, Salutem Ozone, VetOzone Medical Ozone Device, Izmir, Türkiye) twice a week by IP route in addition to the BEP regimen. Medical ozone was applied at a dose of 1 mg/kg, adjusted based on the body weight of each rat [42,43]. In the medical ozone group, only the specified medical ozone dose was administered. All chemotherapy agents used were prepared in a Class II Biosafety Cabinet (Bilser^®^, Ankara, Türkiye) using special protective clothing and equipment, and 0.9% NaCl was used as a diluent.

### 4.3. Collection and Preparation of Tissue and Samples

At the end of the treatment period, the body weights of all the rats were recorded, and the rats were anesthetized by IP administration of 8–10 mg/kg xylazine (Rompun, CAS: 23076-35-9, Bayer, Leverkusen, Germany) and 80–100 mg/kg ketamine (Ketalar, CAS: 6740-88-1, Pfizer, Lüleburgaz Türkiye). Subsequently, the rats were sacrificed by intracardiac puncture under general anesthesia. The right and left testes and epididymis of the rats were excised and placed in a petri dish on ice. TWI for each animal was calculated by using the following formula [49]:
TWI: [(right testis + left testis weights)/Body weight] × 100

### 4.4. Sperm Collection and Examination

The epididymis was finely minced using a scalpel in 1 mL of Global Medium (CooperSurgical/The LifeGlobal Groupl^®^ Guilford, CT, USA) and incubated at 37 °C for 15 min. Following the incubation period, the liquid part was collected using a Pasteur pipette and transferred to Eppendorf tubes. Sperm samples were diluted in Phosphate Buffered Saline (PBS) at a ratio of 1/10, and 10 µL of the diluted samples were counted in a Makler counting chamber.

For morphological evaluation, the Spermac stain method (FertiPRO, Beernem, West-Flanders, Belgium) was employed in accordance with the manufacturer’s instructions. The samples were subsequently examined under a Zeiss Light Microscope (Zeiss^®^, Oberkochen, Germany). For each rat, 500 sperm were counted and evaluated for head, neck, and tail abnormalities according to the Rat Sperm Morphological Evaluation Guide [50].

The sperm viability ratio was determined using the eosin–nigrosin staining technique, as described previously [51]. For each rat, 200 sperm were enumerated. Eosin penetrates dead or damaged sperm with impaired membrane permeability, whereas nigrosin stains the background, facilitating the identification of cells. Consequently, viable sperm remain unstained, whereas non-viable sperm take up the eosin and appear red or pink [52,53].

### 4.5. Testicular Histopathology

Both testes were fixed in 10% neutral buffered formalin for 72 h and subjected to routine tissue processing. Subsequently, serial sections with a thickness of 2–3 µm were obtained, and hematoxylin and eosin staining (H&E; CAS: 517-28-2, 6359-04-2, respectively, Thermo Fisher Scientific, Waltham, MA, USA) was performed for morphological evaluation and histopathological analysis. A total of at least 30 seminiferous tubules in each section were evaluated under a Zeiss light microscope. The initial seminiferous tubules were randomly selected, while the subsequent tubules were chosen by rotating the slide in a clockwise direction. Histopathological evaluation was performed using the modified Johnsen scoring method [54].

### 4.6. Endocrine and Redox Status Measurement

The blood samples were centrifuged, and the serum was obtained by collecting the supernatant. Serum total testosterone (Elabscience^®^, Wuhan, China), thyroid-stimulating hormone (TSH, Elabscience^®^, Wuhan, China), total antioxidant status (TAS, SunRed^®^, Shanghai, China), and total oxidant status (TOS, SunRed^®^, Shanghai, China) were evaluated using the ELISA method according to the manufacturer’s instructions.

### 4.7. Statistical Analysis

The IBM SPSS Statistics 26 (Statistical Package for the Social Sciences; New York, NY, USA) program was used for all statistical analyses. Descriptive statistical methods such as mean, standard deviation, median, frequency, percentage, minimum, and maximum were used to summarize the general characteristics of the data. The normality of the data was assessed using the Kolmogorov–Smirnov and Shapiro–Wilk tests. Normally distributed quantitative variables were compared using one-way ANOVA, with Tukey’s post hoc test employed for pairwise analyses. Statistical significance was accepted at *p*-values less than 0.05. Bar charts were constructed with GraphPad Prism 8.01 (San Diego, CA, USA).

## 5. Conclusions

The current study demonstrates that medical ozone therapy has beneficial effects against BEP chemotherapy-induced testicular toxicity in rats. Ozone not only preserved sperm quality and viability but also mitigated histological damage, normalized redox homeostasis, and partially restored endocrine parameters such as testosterone and TSH. Importantly, this study is among the first to evaluate repeated-dose ozone therapy during the full course of BEP exposure, reflecting a clinically relevant toxicity model. The outcomes of this experimental investigation may offer valuable preliminary insights into the potential role of medical ozone in preserving testicular structure and function and alleviating oxidative and hormonal disturbances associated with BEP chemotherapy, thereby supporting the rationale for future clinical research.

## Figures and Tables

**Figure 1 ijms-26-08547-f001:**
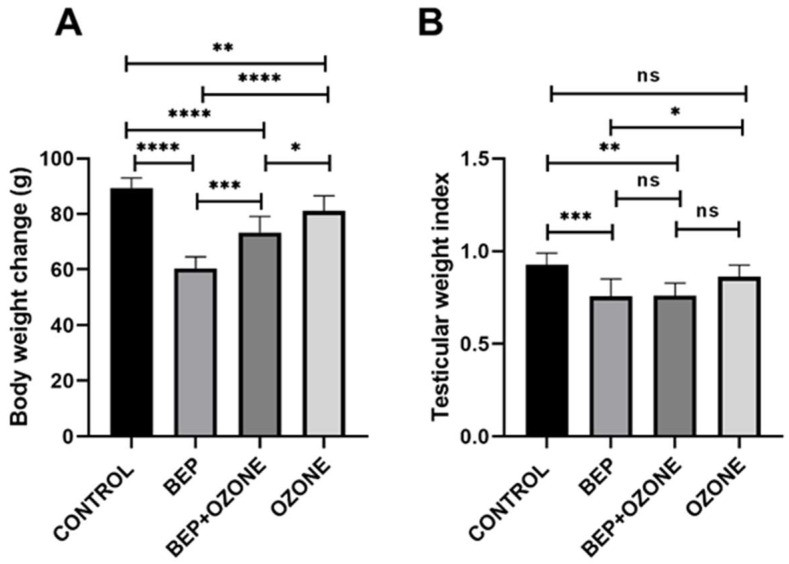
Change in body weight (**A**) and testis weight index (**B**) of (i) control, (ii) bleomycin, etoposide, and cisplatin (BEP)-treated, (iii) BEP- and ozone-treated, and (iv) ozone-alone-treated rats. Data are presented as mean ± SD (* *p* < 0.05, ** *p* < 0.01, *** *p* < 0.001, **** *p* < 0.0001, ns: not significant).

**Figure 2 ijms-26-08547-f002:**
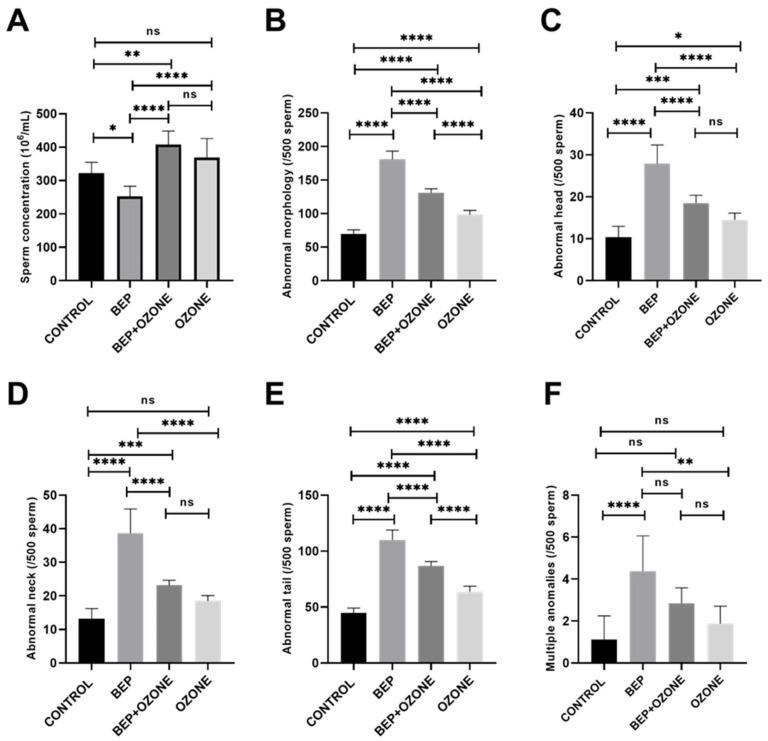
Change in sperm concentration (**A**), abnormal morphology (**B**), abnormal head (**C**), abnormal neck (**D**), abnormal tail (**E**), and multiple anomalies (**F**) of (i) control, (ii) bleomycin, etoposide, and cisplatin (BEP)-treated, (iii) BEP- and ozone-treated, and (iv) ozone-alone-treated rats. Data are presented as mean ± SD (* *p* < 0.05, ** *p* < 0.01, *** *p* < 0.001, **** *p* < 0.0001, ns: not significant).

**Figure 3 ijms-26-08547-f003:**
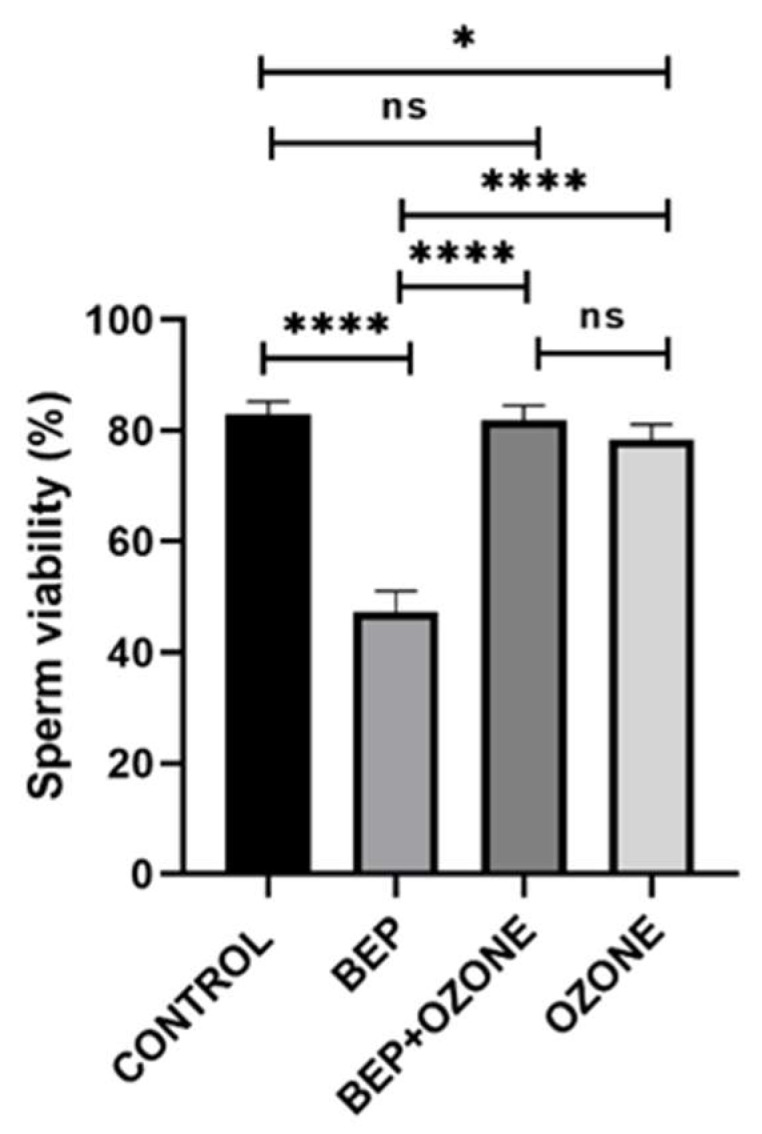
Change in sperm viability of (i) control, (ii) bleomycin, etoposide, and cisplatin (BEP)-treated, (iii) BEP- and ozone-treated, and (iv) ozone-alone-treated rats. Data are presented as mean ± SD (* *p* < 0.05, **** *p* < 0.0001, ns: not significant).

**Figure 4 ijms-26-08547-f004:**
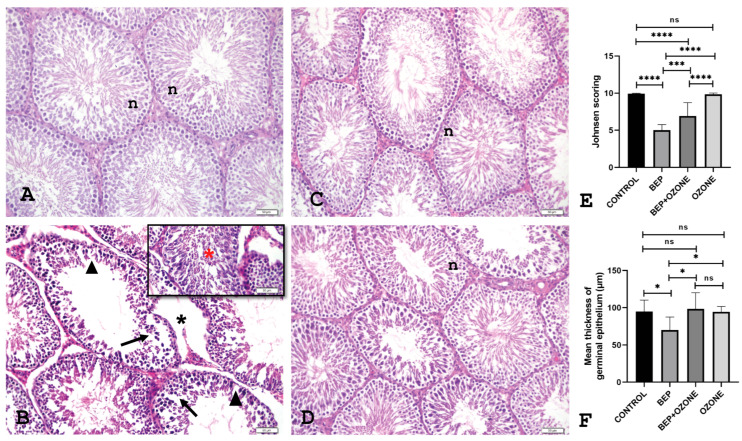
Photomicrographs in the testis sections of control (**A**), bleomycin, etoposide, and cisplatin (BEP)-treated (**B**), BEP- and ozone-treated (**C**), and ozone-alone-treated rats (**D**); n: normal tubule, arrowhead: thinning of the epithelium, arrow: vacuole, black *: loss of integrity of the interstitial space, red *: desquamation. Change in Johnsen scoring (**E**) and the mean thickness of germinal epithelium (**F**) of (i) control, (ii) bleomycin, etoposide, and cisplatin (BEP)-treated, (iii) BEP- and ozone-treated, and (iv) ozone-alone-treated rats. Representative images under (**A**–**D**) 20× magnification. Data are presented as mean ± SD (* *p* < 0.05, *** *p* < 0.001, **** *p* < 0.0001, ns: not significant).

**Figure 5 ijms-26-08547-f005:**
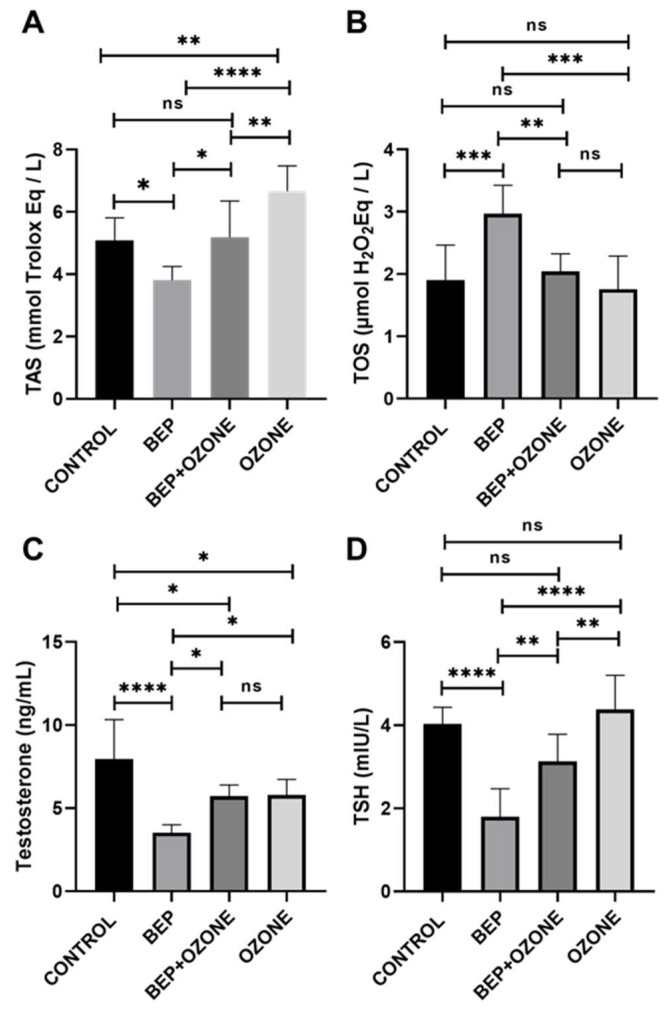
Changes in TAS (**A**), TOS (**B**), testosterone (**C**), and TSH (**D**) levels of (i) control, (ii) bleomycin, etoposide, and cisplatin (BEP)-treated, (iii) BEP- and ozone-treated, and (iv) ozone-only-treated rats. Data are presented as mean ± SD (* *p* < 0.05, ** *p* < 0.01, *** *p* < 0.001, **** *p* < 0.0001, ns: not significant).

**Figure 6 ijms-26-08547-f006:**
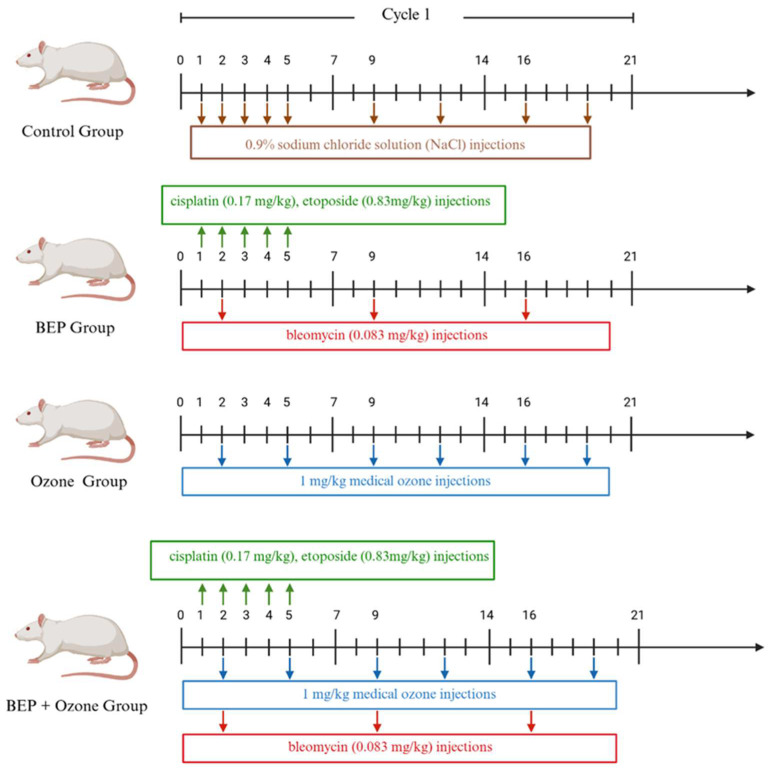
Experimental protocols for the BEP regimen and ozone therapy. Schematic representation of the study design, including group allocation and treatment timeline (created with BioRender.com).

## Data Availability

The data that support the findings of this study are available from the corresponding author upon reasonable request.

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
