# Peer review of "Medical Ozone Treatment Attenuates Male Reproductive Toxicity Induced by Bleomycin, Etoposide, and Cisplatin Regimen in an Experimental Animal Model"

_ijms, 2025, doi:10.3390/ijms26178547_

Round 1
Reviewer 1 Report
Comments and Suggestions for Authors
This study is the first to explore the protective effect of ozone on the reproductive toxicity of BEP chemotherapy. The data indicate that ozone can improve sperm quality, hormone levels and oxidation balance, providing new ideas for clinical protection. But there are still some problems.
line 103 ,Figure 3C-E,it should be Figure 2C-E。
Ozone is also toxic. How did the author choose the optimal dosage and condition?
The article mentioned "32 rats", but did not specify the exact sample size for each group (is it n=8 for each group?).
Figure 3 has no practical significance and should be placed in the supplementary materials. Figure 4 only shows the survival rate and lacks charts such as the deformity rate of the four groups. It is suggested to supplement them. Only 500 sperm were detected under the microscope. These human factors will lead to the results being unfeasible. Why is there no sperm analyzer?
Figure 5 and Figure 6 essentially describe the same thing, with the only difference being quantification. It is suggested that they be combined.
The discussion lacks depth and the mechanism explanation is weak: Only a general mention of "ozone enhancing antioxidant enzymes" is made without associating it with specific molecular pathways (such as the Nrf2/ARE pathway). In the Johnsen scoring of Figure 6, the abnormal sperm rate increased in the ozone-only group, and the reason should be explained in the discussion section. Is it ozone dose-dependent toxicity or the therapeutic window? It is suggested to add an analysis of the feasibility of ozone therapy in clinical application, clearly stating the protective effect and potential limitations of ozone therapy.
It is recommended that detailed information such as the animal ethics number, the model of the ozone instrument, and the tissue processing procedures be described in the Materials and Methods section.
Author Response
Dear Editor,
We would like to thank the reviewers for their careful and thorough evaluation and constructive comments. We tried to address the pointed issues as best as possible. Our responses are described below, and appropriate changes, as suggested by the Reviewers, have been introduced to the manuscript. We would like to thank you very much for your consideration.
Reviewer 1
- line 103 ,Figure 3C-E,it should be Figure 2C-E
Response: We would like to express our gratitude for the Reviewer’s careful review. The figure number has been corrected.
- Ozone is also toxic. How did the author choose the optimal dosage and condition?
Response: We acknowledge the Reviewer’s concern regarding ozone toxicity. Ozone has therapeutic, non-effective, and toxic concentration ranges, and its safety depends on the dose and administration method. For this reason, our study was meticulously designed, supported by an extensive literature review and reference to established guidelines, to determine an appropriate medical ozone dosage in conjunction with the BEP regimen. In the Madrid Declaration on Ozone Therapy, which was formally approved during the "International Meeting of Ozone Therapy Schools" held at the Royal Academy of Medicine in Madrid on June 3–4, 2010, the disease-specific dosage ranges, routes of administration, and mechanisms of action of ozone therapy are clearly defined. Accordingly, ozone concentrations as low as 5–10 µg/mL, and even lower, can exert therapeutic effects while maintaining a wide safety margin. Consequently, a concentration range of 5–60 µg/mL is now widely accepted for both local and systemic applications (https://ozonesociety.org/wp-content/uploads/2016/06/Ozone-Therapy-Madrid_declaration.pdf, Viebahn-Haensler R. & León Fernández O.S., 2024). Similarly, Serra et al. (2023) systematically mapped the evidence on ozone therapy across various medical conditions and administration routes, further emphasizing that the therapeutic use of ozone is supported by a wide safety margin, and toxicity is not observed at these concentrations. We have added this information to the Introduction to ensure that the therapeutic range and safety profile of ozone are clearly presented: “Ozone has therapeutic, non-effective, and toxic concentration ranges, and its safety depends on the dose and administration method. The Madrid Declaration on Ozone Therapy (2010) defines disease-specific dosage ranges and routes, noting that ozone concentrations as low as 5–10 µg/mL, and even lower, can exert therapeutic effects while maintaining a wide safety margin. Similarly, Serra et al. (2023) mapped evidence across medical conditions and administration routes, confirming that therapeutic ozone has a wide safety margin and does not exhibit toxicity within this range.”
In our study, we selected 1 mg/kg intraperitoneal—a dose repeatedly shown to be effective and well tolerated in rat models to ensure safety (Erginel et al., 2023; Simsek et al., 2023). Erginel et al. (2023) and Simsek et al. (2023) were incorporated into the Methods of the manuscript to ensure alignment with previously established experimental protocols and to support the rationale for the selected ozone dosage.
Additionally, a group receiving ozone alone was included to monitor for any toxic effects.
References
-
- https://ozonesociety.org/wp-content/uploads/2016/06/Ozone-Therapy-Madrid_declaration.pdf
- Viebahn-Haensler R, León Fernández OS. Mitochondrial Dysfunction, Its Oxidative Stress-Induced Pathologies and Redox Bioregulation through Low-Dose Medical Ozone: A Systematic Review. Molecules. 2024 Jun 8;29(12):2738. doi: 10.3390/molecules29122738. PMID: 38930804; PMCID: PMC11207058.
- Serra MEG, Baeza-Noci J, Mendes Abdala CV, Luvisotto MM, Bertol CD, Anzolin AP. The role of ozone treatment as integrative medicine. An evidence and gap map. Front Public Health. 2023;10:1112296. doi:10.3389/fpubh.2022.1112296
- Erginel B, Yanar F, Ilhan B, et al. Is the increased ozone dosage key factor for its anti-inflammatory effect in an experimental model of mesenteric ischemia? Ulus Travma Acil Cerrahi Derg. 2023;29(10):1069-1074.
- Simsek EK, Sahinturk F, Gul E, Tepeoglu M, Araz C, Haberal B. Effect of Ozone Therapy on Epidural Fibrosis in Rats. World Neurosurg. 2023;175:e296-e302. doi:10.1016/j.wneu.2023.03.075
- The article mentioned "32 rats", but did not specify the exact sample size for each group (is it n=8 for each group?).
Response: We sincerely thank the esteemed Reviewer for their attention. In the group receiving the combination of BEP and ozone therapy, two rats were excluded from the study in accordance with animal welfare guidelines due to a significant decline in body weight and consequent deterioration in quality of life during the third week of the first treatment cycle. This decision was made in accordance with the recommendation of the attending veterinarian. During the statistical analysis, data from these two animals were excluded from the calculation of any parameters. Consequently, the number of animals in the BEP + ozone group was designated as 6 for all parameters. An explanation is provided in the section titled "4.1. Animals and experimental design" in the manuscript as follows: "Two rats from the group receiving combined BEP and ozone therapy were excluded from the study in the third week of the first cycle due to excessive weight loss and reduced quality of life, based on the recommendations of the attending veterinarian and the animal welfare committee. These animals were excluded from statistical analyses for all parameters."
- Figure 3 has no practical significance and should be placed in the supplementary materials.
Response: We appreciate the Reviewer's suggestions. In accordance, Figure 3 has been removed from the manuscript and submitted as supplementary material.
- Figure 4 only shows the survival rate and lacks charts such as the deformity rate of the four groups. It is suggested to supplement them. Only 500 sperm were detected under the microscope. These human factors will lead to the results being unfeasible. Why is there no sperm analyzer?
Response: We are thankful for the Reviewer's suggestions. Figures 2B-C-D-E-F present the bar graphs that demonstrate the sperm morphological abnormalities, including abnormal morphology, abnormal head, abnormal neck, abnormal tail, and multiple anomalies, in detail.
Regrettably, the laboratory is not equipped with a sperm analyzer. Nevertheless, we used the Eosin-Nigrosin staining technique, a widely employed and indispensable method in research for evaluating sperm viability. This method is characterized by its simplicity, reliability, and capacity to distinctly differentiate live from dead spermatozoa. Furthermore, it is utilized in IVF clinics to assess sperm quality prior to assisted reproductive procedures (Kanna and Shetty, 2023; Agarwal et al., 2021). The one-step eosin-nigrosin technique does not require negative phase contrast optics. Due to its streamlined methodology, it is regarded as a favorable option in terms of standardization and quality control management (Björndahl et al., 2003). A review of the current literature was conducted in order to determine the number of sperm to be examined in the study. Following this analysis, it was decided that 500 would be the optimal cell count (Adamkovicova et al., 2016, Olatunji et al., 2023).
References
-
- Kanna, Sandhyarani, and Archana Shetty. "Eosin nigrosin staining technique in assessment of sperm vitality in medical laboratories–A snippet from our experience on implementing the staining, interpretation and quality control procedures." Indian J Obstetr Gynecol Res 10.2 (2023): 227-229.
- Agarwal A, Sharma RK, Gupta S, et al. Sperm Vitality and Necrozoospermia: Diagnosis, Management, and Results of a Global Survey of Clinical Practice. World J Mens Health. 2022;40(2):228-242. doi:10.5534/wjmh.210149
- Björndahl L, Söderlund I, Kvist U. Evaluation of the one-step eosin-nigrosin staining technique for human sperm vitality assessment. Hum Reprod. 2003;18(4):813-816. doi:10.1093/humrep/deg199
- Adamkovicova, M., Toman, R., Martiniakova, M. et al. Sperm motility and morphology changes in rats exposed to cadmium and diazinon. Reprod Biol Endocrinol 14, 42 (2016). https://doi.org/10.1186/s12958-016-0177-6
- Olatunji AO, Ayo JO, Suleiman MM, Ambali SF, Shittu M, Akorede GJ, Aremu A, Lamidi IY, Afisu B, Adenubi OT. Ameliorative potentials of diosmin and hesperidin fractions on chlorpyriphos-induced changes in reproductive hormones, sperm characteristics, and testicular glycogen in male Wistar rats. Naunyn Schmiedebergs Arch Pharmacol. 2024 Nov;397(11):9181-9190. doi: 10.1007/s00210-024-03241-1. Epub 2024 Jun 21. PMID: 38904770
- Figure 5 and Figure 6 essentially describe the same thing, with the only difference being quantification. It is suggested that they be combined.
Response: In accordance with the Reviewer's suggestion, Figures 5 and 6 have been combined.
- The discussion lacks depth and the mechanism explanation is weak: Only a general mention of "ozone enhancing antioxidant enzymes" is made without associating it with specific molecular pathways (such as the Nrf2/ARE pathway).
Response: We appreciate the Reviewer's insightful comment regarding the need for a more detailed mechanistic explanation of ozone therapy. In response, we have expanded the Introduction to include its ability to induce controlled oxidative stress and simultaneously activate protective antioxidant mechanisms: “Medical ozone (O₃) is a potent oxidizing agent utilized in medical applications at low and controlled doses. Following administration, ozone rapidly dissolves in the body, generating reactive oxygen species (ROS) and lipid peroxidation products (LOPs). These molecules create a controlled oxidative stress environment within cells, thereby stimulating antioxidant defense mechanisms. Consequently, cells exhibit increased resistance to oxidative stress, while molecular responses essential for tissue repair and inflammation regulation are also activated (Sagai and Bocci, 2011; Franzini et al., 2022).”
We also expanded Discussion to include specific molecular pathways activated by ozone, particularly focusing on the Keap1-Nrf2-ARE signaling pathway, which plays a pivotal role in mediating the therapeutic effects of ozone: “Mild or low-dose medical ozone therapy exerts its molecular effects primarily through controlled oxidative stress that activates adaptive cellular responses. This oxidative stimulus enhances the antioxidant defense system by upregulating key pathways such as nuclear factor erythroid 2-related factor 2/ Kelch-like ECH-associated protein (Nrf2/Keap1), leading to increased expression of phase II detoxifying enzymes and antioxidant molecules, including SOD, CAT, and GSH-Px (Franzini et al., 2023). Additionally, ozone-induced mild oxidative stress promotes the release of growth factors and modulates inflammatory signaling by activating Nrf2, while concurrently inhibiting NF-kB, thereby reducing the production of proinflammatory cytokines and favoring an anti-inflammatory phenotype (Jeyaraman et al. 2024). When activated under oxidative stress, Nrf2 not only enhances antioxidant and antiinflammatory defenses but also confers anti-apoptotic effects and supports mitochondrial function and cellular bioenergetics (Malatesta et al. 2024, Galiè et al. 2019). Collectively, these mechanisms suggest that controlled ozone exposure triggers adaptive responses that restore redox homeostasis and support tissue repair.”
References
-
- Sagai, M. and Bocci, V. (2011) Mechanisms of Action Involved in Ozone Therapy: Is Healing Induced via a Mild Oxidative Stress? Medical Gas Research, 1, Article No. 29. doi:10.1186/2045-9912-1-29
- Franzini M, Valdenassi L, Pandolfi S, Tirelli U, Ricevuti G, Simonetti V, Berretta M, Vaiano F, Chirumbolo S. The biological activity of medical ozone in the hormetic range and the role of full expertise professionals. Front Public Health. 2022 Sep 16;10:979076. doi: 10.3389/fpubh.2022.979076.
- Franzini M, Valdenassi L, Pandolfi S, Tirelli U, Ricevuti G, Chirumbolo S. The Role of Ozone as an Nrf2-Keap1-ARE Activator in the Anti-Microbial Activity and Immunity Modulation of Infected Wounds. Antioxidants (Basel). 2023 Nov 8;12(11):1985. doi: 10.3390/antiox12111985. PMID: 38001838; PMCID: PMC10669564.
- Jeyaraman M, Jeyaraman N, Ramasubramanian S, Balaji S, Nallakumarasamy A, Patro BP, Migliorini F. Ozone therapy in musculoskeletal medicine: a comprehensive review. Eur J Med Res. 2024 Jul 31;29(1):398. doi: 10.1186/s40001-024-01976-4. PMID: 39085932; PMCID: PMC11290204.
- Malatesta, M.; Tabaracci, G.; Pellicciari, C. Low-Dose Ozone as a Eustress Inducer: Experimental Evidence of the Molecular Mechanisms Accounting for Its Therapeutic Action. Int. J. Mol. Sci. 2024, 25, 12657. https://doi.org/10.3390/ijms252312657
- Galiè M, Covi V, Tabaracci G, Malatesta M. The Role of Nrf2 in the Antioxidant Cellular Response to Medical Ozone Exposure. Int J Mol Sci. 2019 Aug 17;20(16):4009. doi: 10.3390/ijms20164009. PMID: 31426459; PMCID: PMC6720777.
- In the Johnsen scoring of Figure 6, the abnormal sperm rate increased in the ozone-only group, and the reason should be explained in the discussion section. Is it ozone dose-dependent toxicity or the therapeutic window? It is suggested to add an analysis of the feasibility of ozone therapy in clinical application, clearly stating the protective effect and potential limitations of ozone therapy.
Response: We thank the reviewer for his/her comment. However, in the graphs for Johnsen score and mean thickness of the germinal epithelium, only the ozone-only group shows no statistically significant difference compared to the control. A statistically significant difference is observed in the percentage of abnormal sperm morphology. Accordingly, we have added the following explanation to the Discussion section to clarify this point:
“Nevertheless, the slight increase in sperm abnormalities observed in the ozone-only group indicates that the toxic dose may vary in healthy animals and that optimal dosing differs from disease models. Its application in healthy rats without underlying oxidative stress may lead to an imbalance between ROS and antioxidant defenses. This imbalance can result in oxidative damage to spermatozoa, manifesting as morphological abnormalities. Previous studies have reported that exposure to environmental ozone adversely affects semen quality, including sperm morphology, in humans. It has been demonstrated that the disruption of the antioxidant-oxidant balance in favor of oxidants is a prerequisite for the efficacy of damage prevention (Sokol et al., 2006, Jedlinska-Krakowska et al., 2006, Tusat et al, 2017).”
References
-
- Sokol RZ, Kraft P, Fowler IM, Mamet R, Kim E, Berhane KT. Exposure to environmental ozone alters semen quality. Environ Health Perspect. 2006 Mar;114(3):360-5. doi: 10.1289/ehp.8232. PMID: 16507458; PMCID: PMC1392229.
- Jedlinska-Krakowska M, Bomba G, Jakubowski K, Rotkiewicz T, Jana B, Penkowski A. Impact of oxidative stress and supplementation with vitamins E and C on testes morphology in rats. J Reprod Dev. 2006 Apr;52(2):203-9. doi: 10.1262/jrd.17028. Epub 2005 Dec 28. PMID: 16394623
- Tusat M, Mentese A, Demir S, Alver A, Imamoglu M. Medical ozone therapy reduces oxidative stress and testicular damage in an experimental model of testicular torsion in rats. Int Braz J Urol. 2017 Nov-Dec;43(6):1160-1166. doi: 10.1590/S1677-5538.IBJU.2016.0546. PMID: 28727368; PMCID: PMC5734081
- It is recommended that detailed information such as the animal ethics number, the model of the ozone instrument, and the tissue processing procedures be described in the Materials and Methods section.
Response: The ethical committee approval obtained from Maltepe University is provided under the headings “4.1. Animals and experimental design” and “Institutional Review Board Statement. “The experimental protocol adhered to international guidelines for the care and use of laboratory animals and was approved by the Maltepe University Local Ethics Committee for Animal Experiments (protocol number 2022.09.01).”
The ozone instrument, its brand, and supplier information are specified in section 4.2, 'Experimental protocols for the BEP regimen and ozone therapy. “The BEP and medical ozone group was administered with medical ozone (Has Medical®, Salutem Ozone, VetOzone Medical Ozone Device, Izmir, Türkiye) twice a week by IP route in addition to the BEP regimen.”
Since the procedures for tissue collection and histological tissue processing are routine and well-established methods that are widely known in the field, detailed explanations have not been included herein.
Johnsen scoring is a well-established and widely used method. Therefore, to avoid overloading the manuscript with excessive details, a detailed description was not included, but a citation was provided.
Reviewer 2 Report
Comments and Suggestions for Authors
Dear authors, the work you have conceived is interesting and certainly offers many points to reflect on. This manuscript addresses a relevant and timely topic: the potential of medical ozone therapy to attenuate male reproductive toxicity induced by the bleomycin–etoposide–cisplatin (BEP) chemotherapy regimen. The main novelty lies in administering ozone treatment throughout the full 9-week BEP protocol, mimicking a clinically relevant exposure scenario. The study is well-structured, with a clear experimental design, adequate group sizes (n=8 per group), and a comprehensive set of outcome measures (sperm parameters, histology, endocrine markers, and oxidative stress indices). However, there are methodological and interpretative issues that reduce the strength of the conclusions.
-
Lack of dose–response analysis – Only one ozone dose (1 mg/kg IP) is tested, with no rationale provided for its selection and no comparison to other doses reported in the literature.
- Justify the dose and administration schedule of ozone with reference to dose-finding studies or clinical analogues.
-
Overinterpretation of results – In several parameters (e.g., testosterone, sperm morphology), ozone leads to only partial recovery, yet the text often describes these as “restored” to control levels.
-
Reframe the conclusions to be more cautious and proportionate to the data.
-
Update the discussion with recent literature on both the beneficial and potentially harmful effects of ozone in non-pathological conditions.
-
-
Ozone-only control effects – The ozone-alone group shows negative changes (increased sperm head/tail abnormalities, slight reduction in viability), but this is not discussed in detail. This raises safety concerns that must be addressed.
-
Statistical issues – The manuscript does not report testing for homogeneity of variance; Tukey’s post hoc test is mentioned even after Kruskal–Wallis, which is inappropriate for non-parametric data.
- Review and, if necessary, correct the statistical analysis, using appropriate post hoc tests for non-parametric data.
-
Translational relevance – The discussion claims clinical applicability but does not sufficiently address pharmacokinetic and physiological differences between rats and humans, particularly for intraperitoneal ozone administration.
-
Provide a critical discussion of the negative effects observed in the ozone-only group and possible mechanisms (e.g., excessive oxidative challenge, inappropriate dosing).
- Explicitly acknowledge the study limitations – absence of fertility trials, lack of long-term follow-up, and use of a single animal model.
-
-
Literature gaps – Some cited references are only tangentially related (e.g., ischemia–reperfusion models), and the discussion omits recent work on the potential pro-oxidant effects of ozone at higher doses.
Author Response
Dear Editor,
We would like to thank the reviewers for their careful and thorough evaluation and constructive comments. We tried to address the pointed issues as best as possible. Our responses are described below, and appropriate changes, as suggested by the Reviewers, have been introduced to the manuscript. We would like to thank you very much for your consideration.
Reviewer 2
- Lack of dose–response analysis – Only one ozone dose (1 mg/kg IP) is tested, with no rationale provided for its selection and no comparison to other doses reported in the literature. Justify the dose and administration schedule of ozone with reference to dose-finding studies or clinical analogues.
Response: We thank the esteemed reviewer for their valuable suggestions and for raising important points regarding the doses and potential toxicity of ozone. It is well established that ozone has therapeutic, non-effective, and toxic concentration ranges; therefore, careful dose selection is critical. In planning this study, we conducted an extensive literature review and consulted relevant guidelines to determine the most appropriate medical ozone dosage in conjunction with the BEP regimen.
The Madrid Declaration on Ozone Therapy, approved during the “International Meeting of Ozone Therapy Schools” held at the Royal Academy of Medicine in Madrid (June 3–4, 2010) under the auspices of the Spanish Association of Medical Professionals in Ozone Therapy, clearly defines disease-specific dosage ranges, administration routes, and mechanisms of action for ozone therapy. According to this declaration, concentrations as low as 5–10 µg/mL have therapeutic effects with a wide safety margin, and the generally accepted therapeutic range extends from 5 to 60 µg/mL for both local and systemic applications (https://ozonesociety.org/wp-content/uploads/2016/06/Ozone-Therapy-Madrid_declaration.pdf). The administration schedule of medical ozone was designed based on that of the BEP regimen to minimize its detrimental effects.
Following this guidance and previous experimental work, the ozone administration dose in our rat model was selected to align with the doses used in recent studies by Erginel et al. (2023) and Simsek et al. (2023), corresponding to 1 mg/kg intraperitoneally (approximately 50 µg/mL). This dose has been shown to represent the highest therapeutic concentration without inducing toxicity, while doses exceeding 60 µg/mL may present toxic potential (Viebahn-Haensler & León Fernández, 2024; https://ozonesociety.org/wp-content/uploads/2016/06/Ozone-Therapy-Madrid_declaration.pdf).
Supporting evidence from multiple experimental models reinforces the safety and efficacy of this dosing regimen. In radiation-induced liver and small intestinal injury, 1 mg/kg IP ozone reduced malondialdehyde (MDA) levels, enhanced antioxidant enzyme activities (SOD, CAT, GSH), and improved histopathology (Gultekin et al., 2013). In a frozen shoulder model, intra-articular ozone at the same dose decreased inflammation and promoted functional recovery, suggesting therapeutic potential comparable to corticosteroids (Simsek et al., 2025). In gastrointestinal mucosa, 1 mg/kg IP ozone increased villus height and crypt depth, supporting epithelial proliferation and homeostasis (Sukhotnik et al., 2015). Similarly, in a model of intra-abdominal infection, this dose reduced systemic inflammatory markers (IL-6, CINC-1) (Souza et al., 2010). To further account for potential species-specific or experimental-condition-related effects, we included a group that received ozone alone to monitor any possible toxic outcomes.
Based on the reviewer’s suggestion, the following text has been added to the Discussion section:
Although medical ozone exhibits protective effects at therapeutic doses, it is important to note that higher concentrations may potentially exert toxic effects, highlighting the need for careful dose selection (Viebahn-Haensler R. ve León Fernández OS., 2024). We selected an ozone dose of 1 mg/kg was selected based on extensive literature evidence and established guidelines (https://ozonesociety.org/wp-content/uploads/2016/06/Ozone-Therapy-Madrid_declaration.pdf, Erginel et al. 2023, Simsek et al., 2025). Additionally, a group receiving ozone alone was included to monitor for any toxic effects.
References
-
- https://ozonesociety.org/wp-content/uploads/2016/06/Ozone-Therapy-Madrid_declaration.pdf
- Viebahn-Haensler R, León Fernández OS. Mitochondrial Dysfunction, Its Oxidative Stress-Induced Pathologies and Redox Bioregulation through Low-Dose Medical Ozone: A Systematic Review. Molecules. 2024 Jun 8;29(12):2738. doi: 10.3390/molecules29122738. PMID: 38930804; PMCID: PMC11207058
- Erginel B, Yanar F, Ilhan B, et al. Is the increased ozone dosage key factor for its anti-inflammatory effect in an experimental model of mesenteric ischemia? Ulus Travma Acil Cerrahi Derg. 2023;29(10):1069-1074.
- Simsek EK, Sahinturk F, Gul E, Tepeoglu M, Araz C, Haberal B. Effect of Ozone Therapy on Epidural Fibrosis in Rats. World Neurosurg. 2023;175:e296-e302. doi:10.1016/j.wneu.2023.03.075
- Gultekin FA, Bakkal BH, Guven B, Tasdoven I, Bektas S, Can M, Comert M. Effects of ozone oxidative preconditioning on radiation-induced organ damage in rats. J Radiat Res. 2013 Jan;54(1):36-44. doi: 10.1093/jrr/rrs073. Epub 2012 Aug 21. PMID: 22915786; PMCID: PMC3534275.
- Sukhotnik I, Starikov A, Coran AG, Pollak Y, Sohotnik R, Shaoul R. Effect of ozone on intestinal epithelial homeostasis in a rat model. Rambam Maimonides Med J. 2015 Jan 29;6(1):e0006. doi: 10.5041/RMMJ.10181. PMID: 25717388; PMCID: PMC4327322
- Souza YM, Fontes B, Martins JO, Sannomiya P, Brito GS, Younes RN, Rasslan S. Evaluation of the effects of ozone therapy in the treatment of intra-abdominal infection in rats. Clinics (Sao Paulo). 2010 Feb;65(2):195-202. doi: 10.1590/S1807-59322010000200012. PMID: 20186304; PMCID: PMC2827707
- Overinterpretation of results – In several parameters (e.g., testosterone, sperm morphology), ozone leads to only partial recovery, yet the text often describes these as “restored” to control levels.
Reframe the conclusions to be more cautious and proportionate to the data.
Response: We would like to thank the reviewer for their valuable comments. We made the revisions accordingly in the Abstract, Results, and Discussion sections and emphasized partial recovery/restoration where applicable.
Update the discussion with recent literature on both the beneficial and potentially harmful effects of ozone in non-pathological conditions.
Response: A detailed review has been conducted on the recent use of medical ozone in both clinical and experimental models, including its dosing and routes of administration, and has been added to the Introduction section as suggested by the Reviewer:
“Ozone has therapeutic, non-effective, and toxic concentration ranges, and its safety depends on the dose and administration method. The Madrid Declaration on Ozone Therapy (2010) defines disease-specific dosage ranges and routes, noting that ozone concentrations as low as 5–10 µg/mL, and even lower, can exert therapeutic effects while maintaining a wide safety margin (The Madrid Declaration on Ozone Therapy (2010)). Similarly, Serra et al. (2023) mapped evidence across medical conditions and administration routes, confirming that therapeutic ozone has a wide safety margin and does not exhibit toxicity within this range (Serra et al, 2023).”
- Ozone-only control effects – The ozone-alone group shows negative changes (increased sperm head/tail abnormalities, slight reduction in viability), but this is not discussed in detail. This raises safety concerns that must be addressed.
Response: We acknowledge the Reviewer’s concern regarding the negative changes in the ozone-only group. The Madrid Declaration on Ozone Therapy, approved at the “International Meeting of Ozone Therapy Schools” held at the Royal Academy of Medicine in Madrid, clearly defines disease-specific dosage ranges, administration routes, and the mechanisms of action of ozone therapy. It has also been noted that doses exceeding the established therapeutic range can overload the antioxidant defense system and increase oxidative damage (Bocci, 2006). Since medical ozone dosing varies depending on the specific condition being treated, the slight increase in certain anomalies observed only in the ozone-only group, compared to the control group, can be explained by the fact that in healthy animals—especially when ozone is applied for preventive rather than therapeutic purposes—the effective dose may differ from that used in disease models.
We have added the following paragraph to the Discussion section as a possible explanation:
“Nevertheless, the slight increase in sperm abnormalities observed in the ozone-only group indicates that the toxic dose may vary in healthy animals and that optimal dosing differs from disease models. Its application in healthy rats without underlying oxidative stress may lead to an imbalance between ROS and antioxidant defenses. This imbalance can result in oxidative damage to spermatozoa, manifesting as morphological abnormalities. Previous studies have reported that exposure to environmental ozone adversely affects semen quality, including sperm morphology, in humans. It has been demonstrated that the disruption of the antioxidant-oxidant balance in favor of oxidants is a prerequisite for the efficacy of damage prevention (Sokol et al., 2006, Jedlinska-Krakowska et al., 2006, Tusat et al, 2017).”
References
-
- https://ozonesociety.org/wp-content/uploads/2016/06/Ozone-Therapy-Madrid_declaration.pdf
- Bocci VA. Scientific and medical aspects of ozone therapy. State of the art. Arch Med Res. 2006, 37(4):425–35.
- Sokol RZ, Kraft P, Fowler IM, Mamet R, Kim E, Berhane KT. Exposure to environmental ozone alters semen quality. Environ Health Perspect. 2006 Mar;114(3):360-5. doi: 10.1289/ehp.8232. PMID: 16507458; PMCID: PMC1392229.
- Jedlinska-Krakowska M, Bomba G, Jakubowski K, Rotkiewicz T, Jana B, Penkowski A. Impact of oxidative stress and supplementation with vitamins E and C on testes morphology in rats. J Reprod Dev. 2006 Apr;52(2):203-9. doi: 10.1262/jrd.17028. Epub 2005 Dec 28. PMID: 16394623
- Tusat M, Mentese A, Demir S, Alver A, Imamoglu M. Medical ozone therapy reduces oxidative stress and testicular damage in an experimental model of testicular torsion in rats. Int Braz J Urol. 2017 Nov-Dec;43(6):1160-1166. doi: 10.1590/S1677-5538.IBJU.2016.0546. PMID: 28727368; PMCID: PMC5734081
- Statistical issues – The manuscript does not report testing for homogeneity of variance; Tukey’s post hoc test is mentioned even after Kruskal–Wallis, which is inappropriate for non-parametric data. Review and, if necessary, correct the statistical analysis, using appropriate post hoc tests for non-parametric data.
Response: We thank the reviewer for his/her attention and valuable remark. The normality of the data was assessed using the Kolmogorov-Smirnov and Shapiro-Wilk tests. All data were normally distributed and as the Reviewer indicated, One Way ANOVA test with the post-hoc Tukey test was applied. The indication of Kruskal–Wallis test was indeed a clerical error in the manuscript.
- Translational relevance – The discussion claims clinical applicability but does not sufficiently address pharmacokinetic and physiological differences between rats and humans, particularly for intraperitoneal ozone administration.
Provide a critical discussion of the negative effects observed in the ozone-only group and possible mechanisms (e.g., excessive oxidative challenge, inappropriate dosing).
Explicitly acknowledge the study limitations – absence of fertility trials, lack of long-term follow-up, and use of a single animal model.
Response: In our study, the effects of intraperitoneally administered medical ozone on chemotherapy-induced gonadotoxicity were investigated. However, as the Reviewer pointed out, for translational applicability, it is essential to critically consider the pharmacokinetic and physiological differences between rats and humans. Accordingly, to address this concern and also indicate the limitations of the study, we added the following paragraph to the Discussion:
“Moreover, it is important to note that our findings were obtained in a rat model with IP ozone administration, and the pharmacokinetics, tissue distribution, and physiological responses may differ substantially in humans. In clinical practice, ozone is administered via the intravenous route or via local administration, which may affect its systemic and local effects. Further limitations include that infertile animals and animals with neoplasia were not represented in the study population. Therefore, while our results provide mechanistic insights into the protective potential of ozone against chemotherapy-induced oxidative damage, further studies are necessary to determine optimal dosing, administration routes, and safety profiles for translational application in humans.”
We also provided a critical discussion on the negative effects of ozone in the Discussion:
“Nevertheless, the slight increase in sperm abnormalities observed in the ozone-only group indicates that the toxic dose may vary in healthy animals and that optimal dosing differs from disease models. Its application in healthy rats without underlying oxidative stress may lead to an imbalance between ROS and antioxidant defenses. This imbalance can result in oxidative damage to spermatozoa, manifesting as morphological abnormalities. Previous studies have reported that exposure to environmental ozone adversely affects semen quality, including sperm morphology, in humans. It has been demonstrated that the disruption of the antioxidant-oxidant balance in favor of oxidants is a prerequisite for the efficacy of damage prevention (Sokol et al., 2006, Jedlinska-Krakowska et al., 2006, Tusat et al, 2017).”
References
-
- Sokol RZ, Kraft P, Fowler IM, Mamet R, Kim E, Berhane KT. Exposure to environmental ozone alters semen quality. Environ Health Perspect. 2006 Mar;114(3):360-5. doi: 10.1289/ehp.8232. PMID: 16507458; PMCID: PMC1392229.
- Jedlinska-Krakowska M, Bomba G, Jakubowski K, Rotkiewicz T, Jana B, Penkowski A. Impact of oxidative stress and supplementation with vitamins E and C on testes morphology in rats. J Reprod Dev. 2006 Apr;52(2):203-9. doi: 10.1262/jrd.17028. Epub 2005 Dec 28. PMID: 16394623
- Tusat M, Mentese A, Demir S, Alver A, Imamoglu M. Medical ozone therapy reduces oxidative stress and testicular damage in an experimental model of testicular torsion in rats. Int Braz J Urol. 2017 Nov-Dec;43(6):1160-1166. doi: 10.1590/S1677-5538.IBJU.2016.0546. PMID: 28727368; PMCID: PMC5734081
- Literature gaps – Some cited references are only tangentially related (e.g., ischemia–reperfusion models), and the discussion omits recent work on the potential pro-oxidant effects of ozone at higher doses.
Response: We sincerely thank the reviewer for their valuable comments. It is notable that there have been only a limited number of studies conducted, which have combined medical ozone therapy and chemotherapeutics. Our revised manuscript has been updated and refined in accordance with the most recent references. We have 10 citations published in the last two years and twenty six citations published later than 2020.
References
-
- AydoÄŸdu İ, Ekin RG, Yıldız P, MirapoÄŸlu SL, Çay A, AydoÄŸdu YE, et al. Does Ozone Administration Have a Protective Effect Against Cisplatin induced Histological Changes in Rat Testis? J Urol Surg. 2019, 6(1):32–7.
- Moghadam MT, Dadfar R, Khorsandi L. The effects of ozone and melatonin on busulfan-induced testicular damage in mice. JBRA Assist Reprod. 2021, 25(2):176-184.
- Salem EA, Salem NA, Hellstrom WJ. Therapeutic effect of ozone and rutin on adriamycin-induced testicular toxicity in an experimental rat model. Andrologia. 2017, 49(1):e12603.
Reviewer 3 Report
Comments and Suggestions for Authors
- The manuscript revealed the protective effect of medical ozone to “increased antioxidant enzyme activity and modulation of oxidative stress,” but with less direct experimental evidence on the Keap1-Nrf2-ARE pathway or other molecular mechanisms (e.g., antioxidant enzyme protein expression, gene expression). It seems like that the results are insufficient to support a complete mechanistic inference. It is better if the authors further provide some data to supplement the study with molecular assays related to oxidative stress and the antioxidant system (such as SOD, CAT, GSH-Px activity measurements, Nrf2 nuclear translocation analysis).
- The ozone-only group exhibited increased sperm morphological abnormalities, decreased sperm viability, and reduced testosterone, but these findings are only briefly mentioned in the discussion. This is critical for drawing conclusions about the safety of clinical applications. It is better to address the potential reproductive toxicity of ozone itself directly in the discussion and to evaluate its dose–response relationship.
- Ozone (O₃) is a strong oxidant that produces reactive oxygen species (ROS) in tissue. Exposure to ROS associated with particles has been shown to cause DNA damage. In this study, ozone dose (1 mg/kg) is cited from [44] but without exploration of a dose range, making it unclear whether this dose approaches a toxicity threshold. Is this dose suitable for clinical use? It is recommended to include multiple ozone dose groups to assess the protective and toxicological profiles at low, medium, and high doses, thereby avoiding overly definitive conclusions from a single-dose design.
Author Response
Dear Editor,
We would like to thank the reviewers for their careful and thorough evaluation and constructive comments. We tried to address the pointed issues as best as possible. Our responses are described below, and appropriate changes, as suggested by the Reviewers, have been introduced to the manuscript. We would like to thank you very much for your consideration.
Rewiever 3
- The manuscript revealed the protective effect of medical ozone to “increased antioxidant enzyme activity and modulation of oxidative stress,” but with less direct experimental evidence on the Keap1-Nrf2-ARE pathway or other molecular mechanisms (e.g., antioxidant enzyme protein expression, gene expression). It seems like that the results are insufficient to support a complete mechanistic inference. It is better if the authors further provide some data to supplement the study with molecular assays related to oxidative stress and the antioxidant system (such as SOD, CAT, GSH-Px activity measurements, Nrf2 nuclear translocation analysis).
Response: We sincerely thank the reviewer for their valuable comments.
In response, we have expanded the Introduction to include its ability to induce controlled oxidative stress and simultaneously activate protective antioxidant mechanisms: “Medical ozone (O₃) is a potent oxidizing agent utilized in medical applications at low and controlled doses. Following administration, ozone rapidly dissolves in the body, generating reactive oxygen species (ROS) and lipid peroxidation products (LOPs). These molecules create a controlled oxidative stress environment within cells, thereby stimulating antioxidant defense mechanisms. Consequently, cells exhibit increased resistance to oxidative stress, while molecular responses essential for tissue repair and inflammation regulation are also activated (Sagai and Bocci, 2011; Franzini et al., 2022).”
We also expanded Discussion to include specific molecular pathways activated by ozone, particularly focusing on the Keap1-Nrf2-ARE signaling pathway, which plays a pivotal role in mediating the therapeutic effects of ozone: “Mild or low-dose medical ozone therapy exerts its molecular effects primarily through controlled oxidative stress that activates adaptive cellular responses. This oxidative stimulus enhances the antioxidant defense system by upregulating key pathways such as nuclear factor erythroid 2-related factor 2/ Kelch-like ECH-associated protein (Nrf2/Keap1), leading to increased expression of phase II detoxifying enzymes and an-ti-oxidant molecules, including SOD, CAT, and GSH-Px (Franzini et al., 2023). Additionally, ozone-induced mild oxidative stress promotes the release of growth factors and modulates inflammatory signaling by activating Nrf2, while concurrently inhibiting NF-kB, thereby reducing the production of pro-inflammatory cytokines and favoring an anti-inflammatory phenotype (Jeyaraman et al. 2024). When activated under oxidative stress, Nrf2 not only enhances antioxidant and antiinflammatory defenses but also confers anti-apoptotic effects and supports mitochondrial function and cellular bioenergetics (Malatesta et al. 2024, Galiè et al. 2019). Collectively, these mechanisms suggest that controlled ozone exposure triggers adaptive responses that restore redox homeostasis and support tissue repair.”
We appreciate the Reviewer’s valuable suggestion to include additional molecular assays related to oxidative stress and the antioxidant system. We agree that such analyses would provide further mechanistic insight. However, due to current limitations in funding and sample availability, it is not possible for us to perform these additional experiments within the scope of this study. Nevertheless, we believe that the present findings still provide important preliminary evidence and contribute to the understanding of the effects of ozone on chemotherapy-induced gonadotoxicity.
References
-
- Sagai M, Bocci V. Mechanisms of Action Involved in Ozone Therapy: Is healing induced via a mild oxidative stress? Med Gas Res. 2011 Dec 20;1:29. doi: 10.1186/2045-9912-1-29. PMID: 22185664; PMCID: PMC3298518.
- Franzini M, Valdenassi L, Pandolfi S, Tirelli U, Ricevuti G, Simonetti V, Berretta M, Vaiano F, Chirumbolo S. The biological activity of medical ozone in the hormetic range and the role of full expertise professionals. Front Public Health. 2022 Sep 16;10:979076. doi: 10.3389/fpubh.2022.979076. PMID: 36187636; PMCID: PMC9523567
- Franzini M, Valdenassi L, Pandolfi S, Tirelli U, Ricevuti G, Chirumbolo S. The Role of Ozone as an Nrf2-Keap1-ARE Activator in the Anti-Microbial Activity and Immunity Modulation of Infected Wounds. Antioxidants (Basel). 2023 Nov 8;12(11):1985. doi: 10.3390/antiox12111985. PMID: 38001838; PMCID: PMC10669564
- Jeyaraman M, Jeyaraman N, Ramasubramanian S, Balaji S, Nallakumarasamy A, Patro BP, Migliorini F. Ozone therapy in musculoskeletal medicine: a comprehensive review. Eur J Med Res. 2024 Jul 31;29(1):398. doi: 10.1186/s40001-024-01976-4. PMID: 39085932; PMCID: PMC11290204.
- Malatesta, M.; Tabaracci, G.; Pellicciari, C. Low-Dose Ozone as a Eustress Inducer: Experimental Evidence of the Molecular Mechanisms Accounting for Its Therapeutic Action. Int. J. Mol. Sci. 2024, 25, 12657. https://doi.org/10.3390/ijms252312657
- Galiè M, Covi V, Tabaracci G, Malatesta M. The Role of Nrf2 in the Antioxidant Cellular Response to Medical Ozone Exposure. Int J Mol Sci. 2019 Aug 17;20(16):4009. doi: 10.3390/ijms20164009. PMID: 31426459; PMCID: PMC6720777
- The ozone-only group exhibited increased sperm morphological abnormalities, decreased sperm viability, and reduced testosterone, but these findings are only briefly mentioned in the discussion. This is critical for drawing conclusions about the safety of clinical applications. It is better to address the potential reproductive toxicity of ozone itself directly in the discussion and to evaluate its dose–response relationship.
Response: We thank the reviewer for his/her comment. Only the ozone-only group shows no statistically significant difference compared to the control. A statistically significant difference is observed in the percentage of abnormal sperm morphology. Accordingly, we have added the following explanation to the Discussion section to clarify this point:
“Nevertheless, the slight increase in sperm abnormalities observed in the ozone-only group indicates that the toxic dose may vary in healthy animals and that optimal dosing differs from disease models. Its application in healthy rats without underlying oxidative stress may lead to an imbalance between ROS and antioxidant defenses. This imbalance can result in oxidative damage to spermatozoa, manifesting as morphological abnormalities. Previous studies have reported that exposure to environmental ozone adversely affects semen quality, including sperm morphology, in humans. It has been demonstrated that the disruption of the antioxidant-oxidant balance in favor of oxidants is a prerequisite for the efficacy of damage prevention (Sokol et al., 2006, Jedlinska-Krakowska et al., 2006, Tusat et al, 2017).”
References
-
- Sokol RZ, Kraft P, Fowler IM, Mamet R, Kim E, Berhane KT. Exposure to environmental ozone alters semen quality. Environ Health Perspect. 2006 Mar;114(3):360-5. doi: 10.1289/ehp.8232. PMID: 16507458; PMCID: PMC1392229.
- Jedlinska-Krakowska M, Bomba G, Jakubowski K, Rotkiewicz T, Jana B, Penkowski A. Impact of oxidative stress and supplementation with vitamins E and C on testes morphology in rats. J Reprod Dev. 2006 Apr;52(2):203-9. doi: 10.1262/jrd.17028. Epub 2005 Dec 28. PMID: 16394623
- Tusat M, Mentese A, Demir S, Alver A, Imamoglu M. Medical ozone therapy reduces oxidative stress and testicular damage in an experimental model of testicular torsion in rats. Int Braz J Urol. 2017 Nov-Dec;43(6):1160-1166. doi: 10.1590/S1677-5538.IBJU.2016.0546. PMID: 28727368; PMCID: PMC5734081
- Ozone (O₃) is a strong oxidant that produces reactive oxygen species (ROS) in tissue. Exposure to ROS associated with particles has been shown to cause DNA damage. In this study, ozone dose (1 mg/kg) is cited from [44] but without exploration of a dose range, making it unclear whether this dose approaches a toxicity threshold. Is this dose suitable for clinical use? It is recommended to include multiple ozone dose groups to assess the protective and toxicological profiles at low, medium, and high doses, thereby avoiding overly definitive conclusions from a single-dose design.
Response: We acknowledge the Reviewer’s concern regarding ozone toxicity. Ozone has therapeutic, non-effective, and toxic concentration ranges, and its safety depends on the dose and administration method. For this reason, our study was meticulously designed, supported by an extensive literature review and reference to established guidelines, to determine an appropriate medical ozone dosage in conjunction with the BEP regimen. In the Madrid Declaration on Ozone Therapy, which was formally approved during the "International Meeting of Ozone Therapy Schools" held at the Royal Academy of Medicine in Madrid on June 3–4, 2010, the disease-specific dosage ranges, routes of administration, and mechanisms of action of ozone therapy are clearly defined. Accordingly, ozone concentrations as low as 5–10 µg/mL, and even lower, can exert therapeutic effects while maintaining a wide safety margin. Consequently, a concentration range of 5–60 µg/mL is now widely accepted for both local and systemic applications (https://ozonesociety.org/wp-content/uploads/2016/06/Ozone-Therapy-Madrid_declaration.pdf, (Viebahn-Haensler R. & León Fernández O.S., 2024)). Similarly, Serra et al. (2023) systematically mapped the evidence on ozone therapy across various medical conditions and administration routes, further emphasizing that the therapeutic use of ozone is supported by a wide safety margin, and toxicity is not observed at these concentrations. We have added this information to the Introduction to ensure that the therapeutic range and safety profile of ozone are clearly presented: “Ozone has therapeutic, non-effective, and toxic concentration ranges, and its safety depends on the dose and administration method. The Madrid Declaration on Ozone Therapy (2010) defines disease-specific dosage ranges and routes, noting that ozone concentrations as low as 5–10 µg/mL, and even lower, can exert therapeutic effects while maintaining a wide safety margin. Similarly, Serra et al. (2023) mapped evidence across medical conditions and administration routes, confirming that therapeutic ozone has a wide safety margin and does not exhibit toxicity within this range.”
In our study, we selected 1 mg/kg intraperitoneal—a dose repeatedly shown to be effective and well tolerated in rat models to ensure safety (Erginel et al., 2023; Simsek et al., 2023). Erginel et al. (2023) and Simsek et al. (2023) were incorporated into the Methods of the manuscript to ensure alignment with previously established experimental protocols and to support the rationale for the selected ozone dosage.
Additionally, a group receiving ozone alone was included to monitor for any toxic effects.
We would like to thank the reviewer for their valuable comments on overly definitive conclusions. We made the revisions accordingly in the Abstract, Results and Discussion sections and emphasized partial recovery/restoration where applicable.
References
-
- https://ozonesociety.org/wp-content/uploads/2016/06/Ozone-Therapy-Madrid_declaration.pdf)
- Erginel B, Yanar F, Ilhan B, et al. Is the increased ozone dosage key factor for its anti-inflammatory effect in an experimental model of mesenteric ischemia? Ulus Travma Acil Cerrahi Derg. 2023;29(10):1069-1074.
- Simsek EK, Sahinturk F, Gul E, Tepeoglu M, Araz C, Haberal B. Effect of Ozone Therapy on Epidural Fibrosis in Rats. World Neurosurg. 2023;175:e296-e302. doi:10.1016/j.wneu.2023.03.075
- Serra MEG, Baeza-Noci J, Mendes Abdala CV, Luvisotto MM, Bertol CD, Anzolin AP. The role of ozone treatment as integrative medicine. An evidence and gap map. Front Public Health. 2023;10:1112296. doi:10.3389/fpubh.2022.1112296
- Viebahn-Haensler R, León Fernández OS. Mitochondrial Dysfunction, Its Oxidative Stress-Induced Pathologies and Redox Bioregulation through Low-Dose Medical Ozone: A Systematic Review. Molecules. 2024 Jun 8;29(12):2738. doi: 10.3390/molecules29122738. PMID: 38930804; PMCID: PMC11207058.
Round 2
Reviewer 2 Report
Comments and Suggestions for Authors After the revision work, the manuscript has been considerably improved. I have no further suggestions for the authors.Reviewer 3 Report
Comments and Suggestions for Authors
No further comments.